# Label-Free Multi-Microfluidic Immunoassays with Liquid Crystals on Polydimethylsiloxane Biosensing Chips

**DOI:** 10.3390/polym12020395

**Published:** 2020-02-10

**Authors:** Yu-Jui Fan, Fu-Lun Chen, Jian-Chiun Liou, Yu-Wen Huang, Chun-Han Chen, Zi-Yin Hong, Jia-De Lin, Yu-Cheng Hsiao

**Affiliations:** 1School of Biomedical Engineering, Taipei Medical University, 250 Wuxing St., Taipei 11031, Taiwan; ray.yj.fan@tmu.edu.tw (Y.-J.F.); andychen060188@gmail.com (C.-H.C.); b507105053@tmu.edu.tw (Z.-Y.H.); 2International PhD Program for Biomedical Engineering, Taipei Medical University, 250 Wuxing St., Taipei 11031, Taiwan; 3Graduate Institute of Biomedical Optomechatronics, College of Biomedical Engineering, Taipei Medical University, 250 Wuxing St., Taipei 11031, Taiwan; 4Division of Infectious Diseases, Department of Internal Medicine, Wan Fang Hospital, Taipei Medical University, No.111, Sec. 3, Xinglong Rd., Wenshan Dist., Taipei 116, Taiwan; 96003@w.tmu.edu.tw; 5Department of Internal Medicine, School of Medicine, College of Medicine, Taipei Medical University, 250 Wuxing St., Taipei 11031, Taiwan; 6Department of Engineering Science, University of Oxford, Parks Road, Oxford OX1 3PJ, UK; geman1218@yahoo.com.tw

**Keywords:** polydimethylsiloxane, microfluidic, bovine serum albumin

## Abstract

We developed a new format for liquid crystal (LC)-based multi-microfluidic immunoassays, hosted on a polydimethylsiloxane substrate. In this design, the orientations of the LCs were strongly affected by the interface between the four microchannel walls and surrounding LCs. When the alignment layer was coated inside a microchannel, the LCs oriented homeotropically and appeared dark under crossed polarizers. After antigens bound to the immobilized antibodies on the alignment layer were coated onto the channel walls, the light intensity of the LC molecules changed from dark to bright because of disruption of the LCs. By employing pressure-driven flow, binding of the antigen/antibody could be detected by optical signals in a sequential order. The multi-microfluidic LC biosensor was tested by detecting bovine serum albumin (BSA) and an immunocomplex of BSA antigen/antibody pairs, a protein standard commonly used in labs. We show that this multi-microfluidic immunoassay was able to detect BSA and antigen/antibody BSA pairs with a naked-eye detection limitation of −0.01 µg/mL. Based on this new immunoassay design, a simple and robust device for LC-based label-free microfluidic immunodetection was demonstrated.

## 1. Introduction

Microfluidic immunoassays based on polydimethylsiloxane (PDMS) substrates possess several advantages such as short assay times, small volumes, and low costs [1,2]. However, a microfluidic immunoassay needs a special signal detection method and enhancement technique such as fluorescence detection or enzymatic reaction enhancement method, since the miniature size of microfluidic devices reduces the signal. Thus, antibody/antigen pairs in microfluidic devices are usually conjugated with labels like enzymes [3,4], fluorophores [5,6], nanoparticles [7,8], and so on, in order to transduce the immunobinding response into detectable signals. In addition, the biofunctionality of antibody/antigen pairs is affected upon conjugation with labels [9,10].

Recently, liquid crystals (LCs) were successfully used for label-free biodetection [11]. It was reported that the reorientation of LCs caused the immunobinding response to be more sensitive and changed the optical signals of LCs [12,13]. In addition, a protein standard, bovine serum albumin (BSA), was successfully detected using LC materials [14,15]. Such changes in the optical properties of LCs enable unique optical properties such as Bragg reflection, flexibility, and bi-stability [16,17,18,19,20,21]. In addition, the first cholesteric LC (CLC) biosensing device was proposed by Hsiao et al. in 2015 [22]. A highly sensitive color-indicating CLC biosensor was invented. However, it is unfortunate that CLC biosensors require complicated fabrication processes. In addition, LC biosensors must be confined in a well-defined space for biosensing applications such as in a cell device [23] or transmission electron microscopic (TEM) grid [24]. In addition, assembly of these LC devices requires additional complicated steps, which are time-consuming. Currently, these issues are the main obstacles to applying LC biosensors in practice. In addition, Chen et al. invented the first single-substrate CLC biosensing device [25]. It seems that a better way to resolve time-consuming detection by LC biosensors can be found.

Based on the problems mentioned above, we planned to take advantage of the fluidic property of LC materials, as well as propose an LC-based multi-microfluidic immunoassay to detect BSA antigen/antibody pairs. However, some efforts were proposed for integrating LC materials into microfluidic devices. The first LC/PDMS film to detect ethanol in microfluidic devices was recently proposed [26]. The device enables monitoring of ethanol production. However, this device is only applicable for detecting organic molecules. In addition, Liu et al. revealed an LC microfluidic-filled metal grid for detecting detergents and enzymes [27]. Thus, real-time monitoring by an LC microfluidic device has been achieved.

In this study, we developed an LC-based multi-microfluidic immunoassay chip. We show the behavior between BSA antigen/antibody pairs and LC molecules in a PDMS microchannel. We show that this device substantially differs from a typical optical cell. Since the LC molecules within a microchannel are in contact with the four alignment-coated channel walls, the orientations of the LCs are determined by both the surface functionality and the geometric dimensions of the channels. Antigens/antibodies can be detected by observing the optical appearance of LCs inside the microchannel under crossed polarizers. A highly sensitive interface between LC molecules and an alignment layer of N,N-dimethyl-n-octadecyl-3-aminopropyltrimethoxysilyl chloride (DMOAP) was employed to detect BSA concentrations. The schematic of this multi-microfluidic LC/PDMS biosensor is shown in Figure 1.

## 2. Materials and Methods

Single-layer and two-level cascaded microchannels were fabricated by a general PDMS-based soft lithographic fabrication process. First, a microchannel mold was fabricated on a 4-inch (10.16-cm) silicon wafer by standard photolithography using SU-8 2025 photoresist. The thickness of the mold was 25 μm. Then the PDMS base and curing agent were mixed in a 10:1 weight ratio. After allowing 30 min for degassing, the mixture was poured into the master. After baking at 65 °C for 4 h, the cured PDMS microchannel was peeled off the master. Using oxygen plasma treatment, the PDMS microchannel and a pre-cleaned glass substrate were tightly bonded together. The LC mixture used in this study was nematic E44. In addition, the multi-microfluidic channels were immersed in the DMOAP aqueous solution for 30 min to coat the aligned layer onto the inner walls of the microfluidic channel. Next, the multi-microfluidic device was rinsed with deionized water for 1 min in order to remove excess DMOAP solution. For test immobilization, BSA in the form of an aqueous solution was filled into the DMOAP-coated microfluidic channel. In addition, BSA concentrations of 1 and 0.1 mg/mL, and 10, 1, 0.1, and 0.01 µg/mL were used. Concentrations of an anti-BSA antibody of 0, 1, 10, and 100 µg/mL were also used. The empty microfluidic channel was filled with LC under volume flow rates of 5, 10, 20, and 30 µL/min through a syringe to form LC multi-microfluidic chips. A BX51 polarized optical microscope (POM; Olympus, Tokyo, Japan) equipped with a halogen light bulb as the light source was used in this study for observation. All experimental data were acquired at 26 ± 1 °C.

## 3. Results and Discussion

### 3.1. Optical Properties of Multi-Microfluidic LC Biosensors

Polarized optical images of the LC multi-microfluidic biosensors with BSA concentrations of 0 and 1 mg/mL under crossed and paralleled polarizers are shown in Figure 2. One can observe that the darkness and brightness of the microfluidic biosensor corresponded to 0 and 1 mg/mL, respectively. The vertical alignment layer of DMOAP forced the LCs to orient perpendicularly to the surface of the microfluidic channel, and the microfluidic channel without biomolecules appeared dark under crossed polarizers. However, the vertical anchoring force of the alignment layer (DMOAP) was diminished by the immobilized BSA [25], and the brightness of the microfluidic channel increased when observed under crossed polarizers, as shown in Figure 2. Moreover, the phenomenon under parallel polarizers was opposite of that under crossed polarizers. Since the contrast was better under crossed polarizers, we used crossed polarizers for the following quantification measurements. In addition, the LC material used in this report exhibited the LC phase for a wide range of temperatures. Therefore, the LC multi-microfluidic biosensor can conveniently be used in a wide range of temperatures in different settings. Polarized optical images of LC microfluidic biosensors with immobilized BSA protein of 0–1 mg/mL are also shown in Figure 3. In the absence of BSA, the LC multi-microfluidic biosensor was dark, but it became brighter with an increasing BSA concentration. The DMOAP-coated substrate caused LC molecules to be homeotropically aligned. Therefore, it was possible to measure the amount of biomolecules with multi-microfluidic LC chips. In addition, the vertical anchoring force of the alignment layer of DMOAP was diminished by the immobilized BSA layer. The LC molecules converted to a non-homeotropic state with an increasing concentration of BSA. Finally, the planar arrangement caused the biosensor to be in a bright state. In order to achieve quantitative data for LC multi-microfluidic devices, the image from POM was quantified with software [28]. Linear correlations of the intensity of microfluidic LC chips at different BSA concentrations are shown in Figure 4. A higher intensity was shown with a rising BSA concentration. The linear equation between the BSA concentration (*y*) and light intensity (*x*) was *y* = 0.49*x*. These experimental results proved that the LCs in the multi-microfluidic can also be used for quantitating and detecting biomolecules in a linear manner as a new label-free biosensor.

### 3.2. Quantitation for Immunoassay LC Microfluidic Devices

In addition, an immunoassay test of the device was also examined using both BSA and an anti-BSA antibody. Intensities of the immunoassay LC microfluidic devices immobilized with 0, 1, and 10 μg/mL concentrations of BSA and 0, 10, 100, and 1000 µg/mL concentrations of the anti-BSA antibody are shown in Figure 5. We mixed 0–1000 µg/mL of the anti-BSA antibody with identical concentrations of the BSA antigen at concentrations of 0–10 μg/mL to allow the formation of immunocomplexes between specific antigen/antibody pairs. We observed that with lower concentrations of the anti-BSA antibody (<10 µg/mL), immunocomplexes could not form between the specific antigen/antibody pairs. The intensities of the immunocomplexes at 1 and 10 μg/mL concentrations of BSA did not obviously change. When 100 and 1000 μg/mL of the anti-BSA antibody were mixed, the BSA antigen/antibody mixtures produced a much brighter state under POM. However, an excess concentration of the anti-BSA antibody strongly affected the LC arrangement, which meant that the intensity of change in the BSA immunocomplexes could not be analyzed or quantified. These results suggest that BSA immunocomplexes, compared to those with the BSA antigen or antibody alone, induced more significant disruption of the LC arrangement (Figure 5). From the experimental data, 1 µg/mL of the anti-BSA antibody was a more appropriate concentration with the BSA antigen. This method of immunodetection could thus discern between immunocomplexes and unbound antigens and antibodies. The linear correlation between the transmittance intensity of the LC-based multi-microfluidic device and different BSA concentrations of <10 µg/mL of the anti-BSA antibody is shown in Figure 6. We observed that the immunodetection limit of BSA antigen/antibody pairs was 0.01 µg/mL BSA, and that corresponded to 1 µg/mL of the anti-BSA antibody. These results proved that the linear correlation of the LC-based multi-microfluidic device can be used to detect and quantitate biomolecules or for immunodetection in a linear manner. Note that as the antigens and antibodies were complexed through multiple noncovalent interactions, such as hydrogen bonds, electrostatic interactions, van der Waals forces, and so on; 6 h of pre-drying was required to minimize these effects and improve the stability of the BSA immunocomplexes in this study. Based on the sensitivity, label-free state, multi-detection ability, and ease of manufacture, this study shows that LC multi-microfluidic chips have potential for development as a label-free, highly sensitive, cheap, multi-detection, and immunodetection biosensing technique.

### 3.3. Effects of Volume Flow Rates in LC Microfluidic Devices

Volume flow rates in LC microfluidic devices are important because of the fluidity of LC molecules. Figure 7 shows the effects of different volume flow rates of LC injected into the microchannel of chips on the optical performance. Too-rapid volume flow rates (>10 µL/min) resulted in a disordered arrangement of the LCs and produced a defective optical texture [29]. When the volume flow rate was ≥30 µL/min, the too-fast volume flow rates caused a small amount of LC molecules to be left in the microchannel. Due to the disorder of the LCs under rapid volume flow rates, the intensity of the microchannels decreased when the volume flow rate increased. Thus, volume flow rates of <10 µL/min have to be used in LC-based microfluidic devices. In this study, we used a volume flow rate of 5 µL/min to perform LC injection experiments to ensure uniformity of the LCs in the microchannels.

### 3.4. Comparisons to Other Label-Free Biosensing Techniques

Moreover, many label-free biosensing techniques were proposed in the past. The most important and commonly used techniques are grating coupled interferometry (GCI) and plasmonic sensing [30,31,32,33]. The problem with both GCI and plasmonic sensors is that the measurement process is complicated. In addition, both of these sensing devices are expensive and require large equipment, which is not portable or easy to use. Compared to both label-free plasmonic and GCI techniques, our LC microfluidic device is cheaper and can be measured with a smartphone [34] or detailed spectrum. Compared to the well-known immunodetection method, our multi-microfluidic LC immunoassay chips are cheaper and easier to use. Based on the naked-eye detection property of the label-free nature, this study showed that the LC microfluidic device has potential for development as a cheap, sensitive, and portable biosensing technique for immunodetection.

## 4. Conclusions

A novel design of LC-based multi-microfluidic immunoassay chips was proposed. The orientation of the LCs was strongly influenced by the interface between the four channel walls and surrounding LC molecules. When a DMOAP alignment layer was coated onto the interior of the microchannel, the LCs oriented homeotropically and appeared dark under crossed polarizers. After the antigens had bound to the immobilized antibodies in the multi-microfluidic device, the appearance of the LC phase changed from dark to bright because of disruption of the LC orientation. Using pressure-driven flow, antigen/antibody binding could be detected by optical signals in a sequential order. The immunodetection limit of BSA antigen/antibody pairs was 0.01 µg/mL BSA and 1 µg/mL of the anti-BSA antibody. We show that this multi-microfluidic LC immunoassay chip is able to detect BSA and antigen/antibody BSA immunocomplexes with label-free immunodetection. This new design of an immunoassay device provides a sensitive, cheap, multi-detection, and robust approach to LC-based immunodetection.

## Figures and Tables

**Figure 1 polymers-12-00395-f001:**
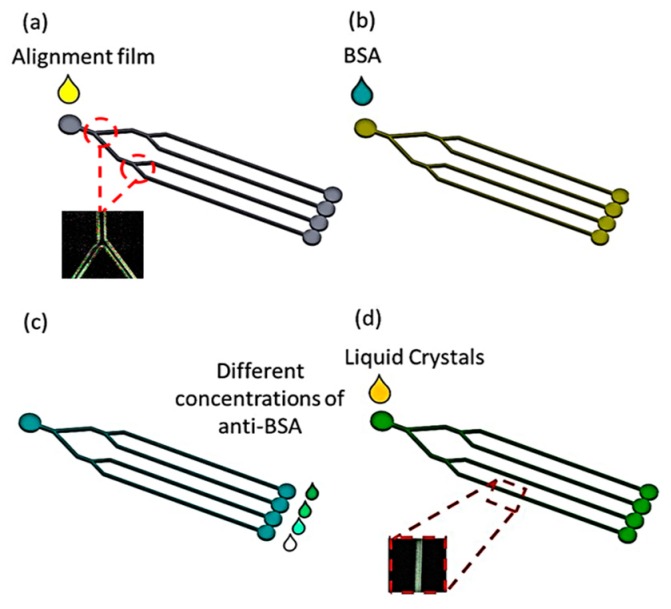
Schematic of multi-microfluidic liquid crystal (LC) immunoassays. The LC configuration changes from the homeotropic to the planar mode in the presence of biomolecules on DMOAP-coated channels. BSA is bovine serum albumin.

**Figure 2 polymers-12-00395-f002:**
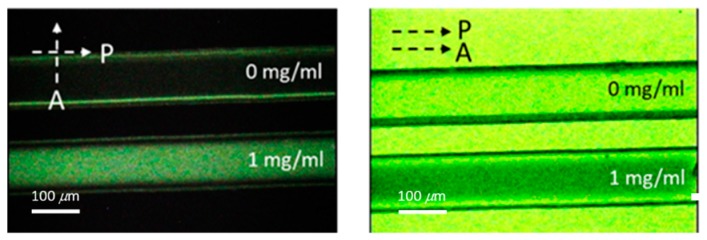
Optical images from a polarized optical microscope of liquid crystal (LC) multi-microfluidic biosensors at both 0 and 1 mg/mL concentrations of bovine serum albumin (BSA) under crossed and parallel polarizer conditions. P stands for polarizer and A is the analyzer.

**Figure 3 polymers-12-00395-f003:**
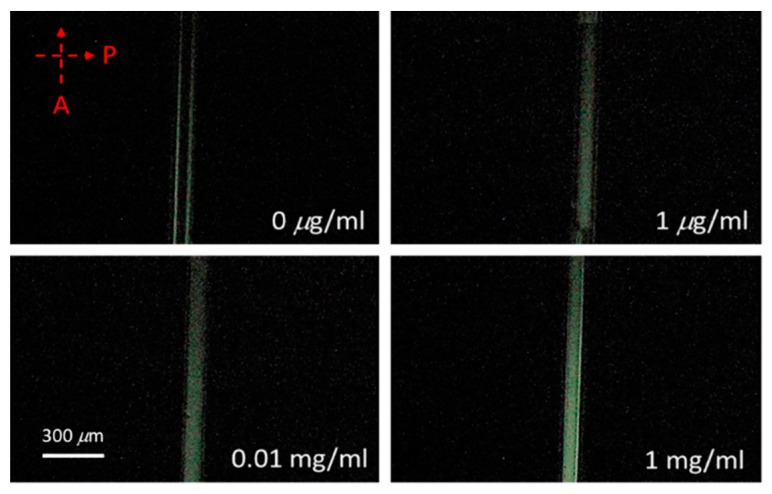
Polarized optical images from a polarized optical microscope of liquid crystal (LC) microfluidic biosensors with immobilized bovine serum albumin (BSA) at 0–1 mg/mL. P stands for polarizer and A is the analyzer.

**Figure 4 polymers-12-00395-f004:**
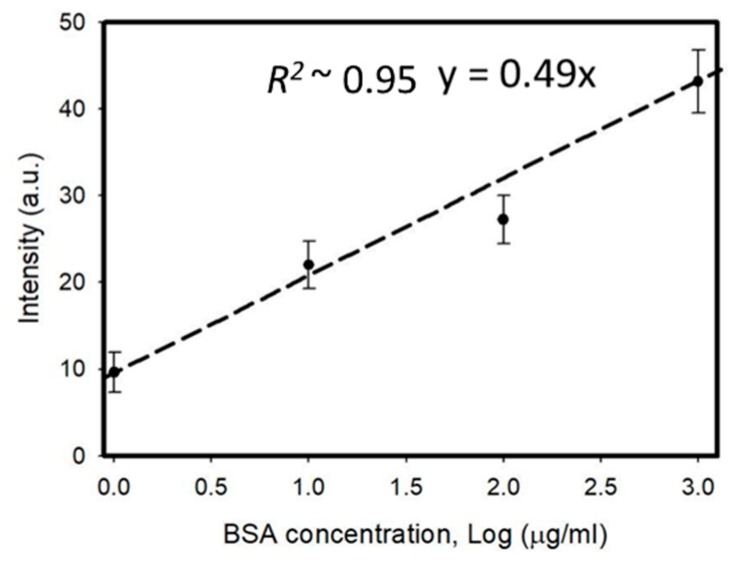
Linear correlations of the transmitted intensity of multi-microfluidic liquid crystal (LC) immunoassay chips at different bovine serum albumin (BSA) concentrations.

**Figure 5 polymers-12-00395-f005:**
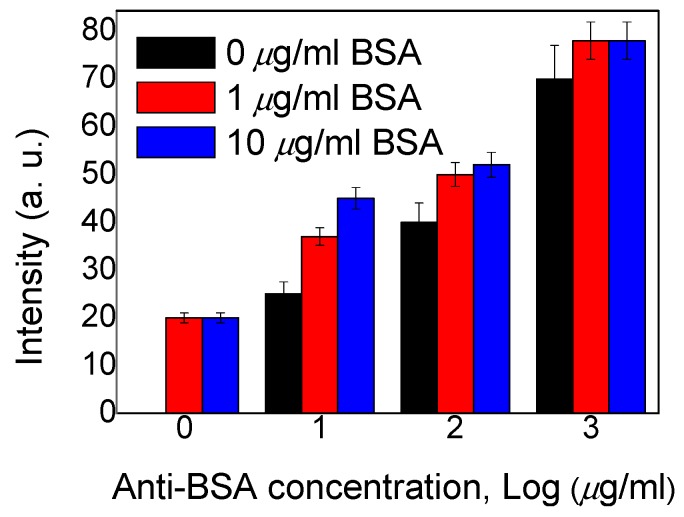
Intensities of immunoassay liquid crystal (LC) microfluidic chips immobilized with 0, 1, and 10 µg/mL concentrations of BSA and 0, 10, 100, and 1000 µg/mL concentrations of the anti-BSA antibody.

**Figure 6 polymers-12-00395-f006:**
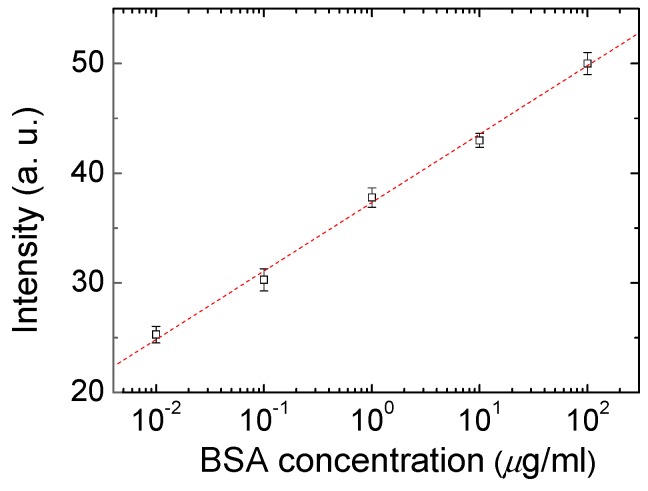
Transmitted intensities of multi-microfluidic liquid crystal (LC) immunoassay chips at different bovine serum albumin (BSA) concentrations mixed with a 10 µg/mL concentration of an anti-BSA antibody.

**Figure 7 polymers-12-00395-f007:**
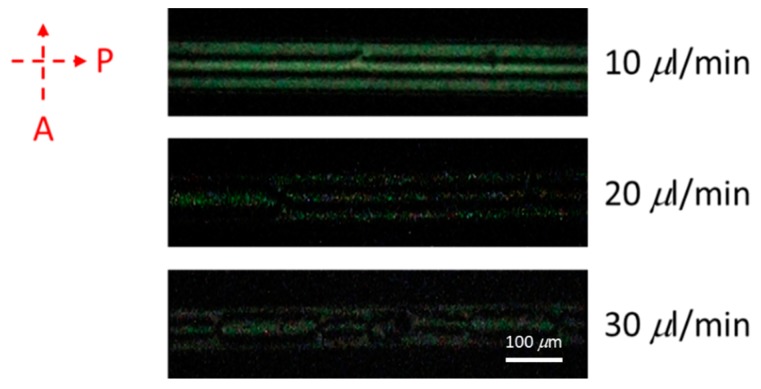
Optical images of a 1 mg/mL concentration of bovine serum albumin (BSA) under different volume flow rates of 10, 20, and 30 µL/min of liquid crystals (LCs) injected into the microchannel. P stands for polarizer and A is the analyzer.

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
