# Peer review of "Label-Free Multi-Microfluidic Immunoassays with Liquid Crystals on Polydimethylsiloxane Biosensing Chips"

_polymers, 2020, doi:10.3390/polym12020395_

Round 1
Reviewer 1 Report
The paper 'Label-free multi-microfluidic immunoassays with liquid crystals on polydimethylsiloxane biosensing chips' presented by Fan Y.-J.and co-authors exploits the property of nematic liquid crystals for the development of a label-free immunoassay test.
The optical properties of the platform were modulated by using BSA as benchmark system. BSA in a concentration lower than 10 ug/ml is capable to disrupt the alignment of liquid crystal. Additionally, the immuno complex with the corresponding antibody can also be detected.
The paper is well written and the conclusions supported by a consistent data set. I recommend the publication in the present form.
Author Response
We thank the reviewer for encouragement.
Reviewer 2 Report
The authors presented a newly developed liquid crystal-based mircofluidic biosensor to to detect albumin. This manuscript suggests a novel technique in terms of label-free microfluidic immunodetection using liquid crystal. However, it seems to be suitable for publication in this journal after minor revisions on the following comments;
1) In Figure 2, scale bar is missing. Please, add a description of the abbreviations A and P to the figure title.
2) In Figure 3, scale bar is missing. Please, add a description of the abbreviations A and P to the figure title.
3) In Figure 4, add P values for a linear regression statistics (y=0.49x).
4) In Figure 5, add SD bars on the bar graph.
5) In Figure 7, scale bar is missing. Please, add a description of the abbreviations A and P to the figure title.
Author Response
We thank the reviewer for good suggestion. In Fig, 2, 3, and 7, we have added a description of the abbreviations A and P to the figure title. In Figure 4, we have added R2 for a linear regression statistic. In Figure 5, we also have added SD bars on the bar graph.
